# Contactless Sensing of Water Properties for Smart Monitoring of Pipelines

**DOI:** 10.3390/s23042075

**Published:** 2023-02-12

**Authors:** Christian Riboldi, Danilo A. Carnevale Castillo, Daniele M. Crafa, Marco Carminati

**Affiliations:** 1Dipartimento di Elettronica, Informazione e Bioingegneria, Politecnico di Milano, 20133 Milano, Italy; 2Istituto Nazionale di Fisica Nucleare (INFN), Sezione di Milano, 20133 Milano, Italy

**Keywords:** ultrasonic flow meter, impedance, lock-in amplifier, wireless sensors networks, internet of things, pipe retrofitting, drinking water, water grid monitoring

## Abstract

A key milestone for the pervasive diffusion of wireless sensing nodes for smart monitoring of water quality and quantity in distribution networks is the simplification of the installation of sensors. To address this aspect, we demonstrate how two basic contactless sensors, such as piezoelectric transducers and strip electrodes (in a longitudinal interdigitated configuration to sense impedance inside and outside of the pipe with potential for impedimetric leak detection), can be easily clamped on plastic pipes to enable the measurement of multiple parameters without contact with the fluid and, thus, preserving the integrity of the pipe. Here we report the measurement of water flow rate (up to 24 m^3^/s) and temperature with ultrasounds and of the pipe filling fraction (capacitance at 1 MHz with ~cm^3^ resolution) and ionic conductivity (resistance at 20 MHz from 700 to 1400 μS/cm) by means of impedance. The equivalent impedance model of the sensor is discussed in detail. Numerical finite-element simulations, carried out to optimize the sensing parameters such as the sensing frequency, confirm the lumped models and are matched by experimental results. In fact, a 6 m long, 30 L demonstration hydraulic loop was built to validate the sensors in realistic conditions (water speed of 1 m/s) monitoring a pipe segment of 0.45 m length and 90 mm diameter (one of the largest ever reported in the literature). Tradeoffs in sensors accuracy, deployment, and fabrication, for instance, adopting single-sided flexible PCBs as electrodes protected by Kapton on the external side and experimentally validated, are discussed as well.

## 1. Introduction

The rapid advancements of diverse and complementary electronic technologies such as long-range, low-power radio protocols, low-power microcontrollers, and sensors, along with energy harvesting solutions, have been enabling a pervasive diffusion of the Internet of Things (IoT) [1,2]. Since water plays a crucial role for life on the planet, the application of IoT technologies to improve the management of water for both drinking [3,4,5] and irrigation [6] use is compelling. One of the main limitations to a large-scale diffusion of sensing nodes for wireless monitoring of water distribution infrastructures is the cost of the individual nodes in the grid. While the cost of the electronics is commonly affordable and scales with the number of produced pieces, the major and uncompressible cost terms are due to calibration and installation. The extent of calibration strongly depends on the measurement, but it could be made automatic and parallelizable, analogously to assembly. Moreover, a consolidating trend in distributed sensing is to relax the requirements on the accuracy of the single sensor, thus skipping detailed and time-consuming calibrations and relying, instead, on data fusion from multiple sensors [7,8]: homogenous or heterogeneous ones, real or virtual (i.e., model-based) ones [9].

Consequently, installation remains the main cost in the deployment of wireless sensor networks (WSN) applied to underground water pipes. In fact, when new pipelines are installed, the incremental cost of sensorized sections of pipes is negligible. Instead, when addressing the application of sensors to already installed networks of pipes, different strategies need to be identified, as happens for retrofitting techniques. One promising and cost-effective approach is the application of external sensors on the walls of the pipes. Although this requires digging, it does not require replacing the pipe and thus interrupting the service. If the sensor footprint is compact, a very small hole can be dug, or it can even fit in already existing inspection manholes.

External sensors, not requiring direct contact with the fluid, are indeed very interesting also for newly produced pipes for multiple reasons: (i) the mechanical properties of the pipe (i.e., its nominal max. pressure) are not altered, (ii) no weak discontinuities and junctions, prone to the creation of leaks, are introduced, and (iii) no contamination of the fluid due to the sensor is possible, being especially relevant for edible fluids.

In this paper, we present an experimental demonstration of the measurement of several water parameters by means of contactless sensors. In particular, we adopt only two classes of sensors that can be externally clamped on plastic pipes: piezoelectric transducers and adhesive band electrodes for impedance sensing. With just two sensors, we can show how it is possible to measure six parameters: flow rate and temperature with ultrasounds, water conductivity, leakage, inner filling status, and, potentially, speed by means of impedance (~1–20 MHz). These technologies are compatible with other noninvasive ones operating at higher frequencies, such as RFID (~100 MHz) for contactless interrogation of the pipe integrity [10] and dielectric identification of water traces in oil (~8 GHz) [11]. 

The novelty of this work is the simultaneous characterization of two contactless and low-cost transducers (piezo and impedance electrodes) applied on plastic pipes of realistic dimensions (90 mm diameter) in which water is traveling at high speed (~1 m/s), i.e., representative of realistic scenarios in drinking water distribution networks. We demonstrate the compatibility and complementarity of piezoelectric and impedance sensing. We show how a single impedance readout circuit, a phase-sensitive demodulator, is able to simultaneously detect the volume fraction of the fluid filling the pipe by a capacitive measurement and its ionic conductivity by a resistive one. 

The installation concept to which we refer in this experimentation is shown in Figure 1. The electronic control unit (ECU) with a local battery is placed in a manhole, granting access for installation and bringing to the surface the antenna for the radio communication and, potentially, a compact solar panel for energetic autonomy of the node. Different from other solutions, this installation type is suitable for any depth of the pipeline. The sensors are applied around the pipe by means of clamps. For example, ultrasonic transducers could be fixed by means of a rigid collar, tightened around the pipe with the proper coupling grease that minimizes discontinuities, where the electrical contacts to the impedance electrodes, glued on the plastic pipe, could be embedded as well.

Recently, we have proposed to use impedance to identify water leakage by means of longitudinal electrodes detecting leaks due to the change in ionic conductivity of the soil surrounding the pipe [12]. Differently from techniques mapping the soil impedance for the surface [13], here we sense a local variation of soil conductivity along the pipe. In this validation, we consider a length of the electrodes of 45 cm for laboratory validation, but this approach could be scaled up to several meters.

The paper is organized as follows: Section 2 describes the modeling of the impedance sensor and the optimization of the sensing frequency by means of finite-element simulations of the spectra in different conditions. Section 3 reports the experimental results of both sensor types, which are discussed in Section 4. Section 5 briefly illustrates the future work, and conclusions are drawn in Section 6.

## 2. Impedance Modeling and Simulations

Flow rate is measured by the difference in the time of flight (ToF) of two ultrasonic waves counter-propagating across the fluid. Clamp-on piezoelectric transducers are applied on the external walls of the pipe at a given angle (around 45°): they transmit and receive the acoustic waves traveling in the fluid through the rigid walls of the pipes. Although sound propagates at a lower speed in plastic (~1 km/s) than in iron (5 km/s), PVC pipes are also suitable for this type of contactless sensing. Extensive literature discusses the rules for optimal placement of transducers on pipes for flow rate measurements [14,15,16], as well as techniques for signal processing, such as cross-correlation methods [17]. Arrays of transducers and beam forming can be used to automatically identify geometrical and liquid parameters for significantly faster installation [18]. Ultrasounds could even be used to transmit power and data along pipes, despite having been demonstrated so far only for short distances [19]. Given the vast literature and the high level of consolidation of ultrasonic techniques, in this section, we focus instead on the analysis of the companion electrical impedance part. 

In order to analyze the expected performance, we need to model the electrical impedance equivalent between the electrodes. In addition to such a model with lumped elements, useful to interpret results, detailed numerical simulations (finite element) were performed to assess the limits of detection. Electrodes can be arranged in a single pair or, more commonly, in multiple strips placed symmetrically and connected in an interdigitated way (Figure 2a) around the circumference to cover the whole pipe surface. The equivalent model is shown in Figure 2b. The impedance is composed of two paths in parallel: an inner one and an outer one. Towards the inside, electrodes are coupled to the fluid through the insulating walls of the plastic pipe, modeled by a capacitance C_PIPE_. C_PIPE_ depends on the thickness of the wall and on its dielectric constant (which for plastics is ε_r_ ~2). At low frequencies, the impedance of this capacitance is very large, and it decreases with frequency. When C_PIPE_ is shunted, the bulk ionic resistance of water R_W_, in series to C_PIPE_, becomes electrically accessible. The value of R_W_ depends linearly on the resistivity of water (which depends on the number of ions) and on the geometry of the electrodes, often called *cell constant*. The electric field between the electrodes polarizes water molecules introducing an additional capacitance C_W_ (with ε_r_ ~80) in parallel to the R_W_ branch. Typically, C_W_ < C_PIPE_ and its effect is to limit the width of the resistive plateau of R_W_, which is shorted by C_W_ at higher frequencies. Correspondingly, the frequency window where R_W_ is easily measured is limited at low frequency by C_PIPE_ and at high frequency by C_W_. The larger the ratio C_PIPE_/C_W_, the wider the plateau where R_W_ can be measured to estimate water conductivity, as performed in other contexts of contactless sensing [20,21].

In parallel to the inner path, there is an outer path of impedance through the material surrounding the pipe, which is here assumed to be earthy soil. The interface impedance between the electrodes and soil can be very complicated. Since a minimum level of humidity is also present in dry soil, we assume the presence of some ions giving origin to an interface capacitance C_DL_ and a bulk soil resistance R_SOIL_. Given the irregular morphology of soil, rather than an ideal capacitor C_DL_ is actually a constant phase element (CPE), i.e., a pseudo capacitance commonly used to model electrochemical interfaces [21]. Since the number of ions of water percolating through the earth and their mobility is low, we consider R_SOIL_ >> R_W_. Analogously to the inner path, also in the outer one, a soil capacitance C_SOIL_ has to be added to account for the dielectric behavior of the surrounding medium.

The number and size of electrodes were chosen as a compromise among competing requirements: (i) maximization of sensitivity to leaks [12], (ii) minimization of the metal area, and (iii) homogenous coverage of the surface. For the pipe of 90 mm diameter, four electrodes connected in alternate couples were selected (Figure 2a). The width of the electrodes is 3 cm, and the spacing between them is 4 cm.

Simulations of this geometry were performed by means of COMSOL with a tridimensional full domain, using standard electromagnetic equations in the AC regime. The axial symmetry could have been leveraged to simulate only one-quarter of the domain [21], but as the simulation was fast, this was not necessary. The first simulation, reported in Figure 3a, considers the pipe section surrounded by air (so the soil branch of the equivalent model is absent). Two conditions are simulated: empty pipe (i.e., filled with air) and full pipe (i.e., filled with water). In the case of an empty pipe, R_W_ tends to be infinite, and the global impedance is purely imaginary (with a capacitance equal to 18 pF). When filled with water, C_W_ increases due to the larger dielectric constant of water (giving a total capacitance of 80 pF), and R_W_ is now finite and given by a water conductivity σ_H2O_ = 1000 μS/cm, which represents the upper bound of typical values for tap water. However, R_W_ is not visible in the magnitude plot since the real part of the impedance is negligible with respect to the imaginary one. Its effect is observable in the phase plot with a peak of ~2° at 20 MHz. A lock-in amplifier (LIA) with in-phase and in-quadrature channels can be used to detect tiny variations of the real part of the impedance and, thus, estimate changes in R_W_. Moreover, to check the dependence of the electrical parameters on the electrode’s length L, two lengths were considered (45 cm and 90 cm). As expected, capacitance scales linearly with L.

In the second set of simulations (Figure 3b), earthy soil around the pipe is added, and two values of water conductivity (700 and 1000 μS/cm) and two conditions of the soil (dry and wet due to a small leak) are considered. The baseline value for R_SOIL_ in dry conditions was evaluated by means of experimental measurements, giving an average value σ_SOIL_ = 180 μS/cm. Instead, the leak is simulated with a wet volume of 0.75 L across the electrodes filled with a σ_LEAK_ = 500 μS/cm, which takes into account the soaking of the earth. This simulation is, of course, more realistic. The magnitude of the impedance spectrum shows the regions expected from the lumped model: capacitive at low frequency, resistive in the middle region, and then capacitive again. As visible, in the region around 5 MHz, the difference between dry and wet soil due to the change of R_SOIL_ is maximum (for both values of internal water conductivity), demonstrating the potential for detection of small leaks by means of impedance tracking.

## 3. Experimental Results

### 3.1. Test Setup

A hydraulic loop, shown in Figure 4, was built in the laboratory to test both sensors. A PVC-A pipe, the Fitt Bluforce DN90 PN16 (with a nominal pressure of 16 bar, external diameter of 90 mm, and wall thickness of 4 mm), was chosen as a reasonable compromise between indoor space and the representativeness of the setup. The length of the loop was determined by the flow rate sensor, which requires about 10 diameters of straight pipe, free of perturbations, before the sensor and five diameters after. Thus, the long arm is 2 m long and the short one about 1 m, for a total length of about 6 m and 30 L of circulating water. Motion is created by an immersion pump (model BIOX 400/12 by Robinson & Sons, Ilford, UK) placed in a 60 L tank. The pump is able to create, in this configuration, a maximum flow rate of 24 m^3^/h, which corresponds to a water speed of ~1 m/s, which is consistent with realistic scenarios in drinking water distribution. Two sets of band electrodes were placed on the longest arm. Piezo transducers were connected to a commercial handheld instrument RIF600P by Riels (Padova, Italy). Impedance measurements, instead, were performed with an LIA, model HF2LI by Zurich Instruments (Zurich, Switzerland) endowed with its input current amplifier (HF2TA), with selectable transimpedance gain and operating up to 50 MHz. The LIA bandwidth was set to 100 Hz providing a good tradeoff between temporal resolution and noise.

Before proceeding with impedance measurements, we checked that the pump would not produce interferences. We measured the irradiated spectrum near the pump in the frequency region of interest from 0.1 to 10 MHz with an SDR receiver (SDRPlay by RS Instruments, Corby, UK) operated as a spectrum analyzer with a loop antenna. No change in the spectrum, down to a noise floor of −110 dBm, was observed. Since the piezo transducers operate at a resonance frequency of ~1 MHz, we also checked with the LIA, spanning the frequency around 1 MHz, that no mechanical crosstalk was produced between the operation of the ultrasounds and simultaneous impedance measurements.

### 3.2. Water Flow Rate

Figure 5 reports an example of measurement of the flow rate. Initially, the pump is off, the flow rate is zero, and the control valve is fully open. Then it is switched on, and the flow rate reaches a maximum of 23 m^3^/h (i.e., 1.2 m/s). At a time of 20 s, the valve is closed to 50%, thus increasing the hydraulic resistance and decreasing the flow rate. Then the pump is turned off again, and the flow reaches zero. A zero-flow error is observed, as expected for instruments based on cross-correlation. It is negligible in our case for network monitoring and not metering to customers, where accuracy in the flow rate of a few % of the full-scale range (typically 5 m/s) is acceptable. Anyway, it could be corrected by changing the echo processing algorithm, adopting, for instance, zero-cross sensing. Overall, ultrasonic flow metering is a standard and consolidated measurement. Novel PZT (Lead Zirconate Titanate) materials could be directly applied on the pipe surface to replace commercial transducers [22]. We only highlight that we observed the detrimental effect of air bubbles on the readings. When particles, foams, or debris are present in the water (unlikely in drinking water), more robust approaches such as the Doppler technique [23] should be adopted.

### 3.3. Water Temperature

Temperature is a very stable parameter in water distribution networks, with slow seasonal variations. However, we have shown that real-time temperature sensing can easily provide interesting information about the flows, for instance, when different adductors bring water from different sources at different temperatures mixed together [4]. In order to measure water temperature, a classical temperature probe should be placed in contact with the liquid since an external probe would suffer significant errors due to the large thermal constant of the pipe walls and the surrounding medium.

Since the speed of sound in water depends on its temperature and pressure, an alternative contactless approach to temperature sensing by means of ultrasound ToF measurement is possible. We experimentally verified the feasibility of this approach. Since changing the temperature of the whole volume of water in the loop would have been impractical, we built a static setup for this specific measurement. We used a pipe made of the same PVC-A but with a larger diameter (200 mm), 8.8 mm wall thickness, and a length of 500 mm. It was closed at the base, and the two piezoelectric transducers were installed in the middle at a distance of 72.9 mm, as indicated by the instrument. An RTD temperature sensor was also placed at an intermediate height between the two transducers but out of the acoustic path. The pipe was filled with tap water. After each measurement, the water temperature was increased by adding a small amount of boiling water. The commercial flow meter provides the ToF values. As reported in Figure 6, for increasing temperature, the ToF decreases with a slope of about −250 ns/°C (linearized around room temperature). Since the electronic circuits measuring the ToF have temporal resolution better than 1 ns, sub-degree thermal resolution can be easily achieved. At constant pressure, the relationship between temperature and the speed of sound is nonlinear. If the water is pure, the Belogol’skii and Sekoyan equation could be used to obtain temperature from pressure and the speed of sound values in a range between 0 °C to 40 °C, which is much wider than our interest [24]. The curve obtained experimentally in Figure 6 is translated downwards by a value equal to 53 m/s with respect to the one obtained with the Belogol’skii and Sekoyan equation at atmospheric pressure. Assuming that in the interval 0–40 °C, the offset value remains constant, the temperature could be acquired via an RTD sensor, together with the speed of sound in a single point of the hydraulic section. In this way, we can use it as calibration in order to allow an accurate estimation of the temperature. Calibration is also crucial due to possible temperature gradients along the pipe.

### 3.4. Filling Status

The most straightforward application of impedance sensing from external electrodes on pipes is the dielectric detection of the composition of the fluid flowing inside. Several studies have reported the use of impedance tomography to detect anomalies in fluids by means of multiple electrodes surrounding a pipe made of insulating material [25,26,27]. Since, in our case, the two fluids that can be found in the pipe (neglecting trace elements that require electrochemical techniques to be detected) are water and air, we show how it is possible to measure the filling fraction of the pipe very easily. As reported in Figure 7a, we have tracked the value of the capacitance between the interdigitated electrodes at 1 MHz. Initially, the pipe is empty (i.e., full of air), and the measured capacitance value is 25 pF. This value is slightly larger than the simulated one (18 pF) since the additional parasitic capacitance due to the connections of the electrodes to the measuring instrument was not considered. Then the pump is turned on, and the pipe is quickly filled with water. As expected, the capacitance increases to 85 pF since the relative dielectric constant of C_W_ increases from 1 to 80. The measured capacitance variation of 60 pF is in good agreement with the simulated one (ΔC = 62 pF). Interestingly, a few seconds after the filling of the pipe, fast drops of the capacitance of a few pF are observed. These are attributed to air bubbles circulating in the loop. In fact, bubbles of ~1 cm^3^ volume have been observed both by eye and by artifacts in ultrasounds. This simple measurement shows the potential to monitor the actual filling status of a pipe segment. Moreover, if electrodes are individually driven and sensed, basic tomographic capabilities can be added to the sensing node. 

This high-pressure setup does not allow partial filling of the pipe. Thus, in order to test the linearity of measurement, we used a combination of two sets of electrodes ❶ and ❷ in Figure 4) of the same length, initially measured individually and then connected in parallel to double the length (90 cm) as reported in Figure 7b. Each measurement was repeated a few times obtaining a repeatability of ±0.2 pF. A difference below 10% between the two sets of electrodes was observed, and this is due to the uncertainty in the manual cutting and placement of electrodes and the different lengths (parasitics) of the connection wires from the electrodes to the LIA. As expected, when summing the individual contributions of capacitance, their sum matches the capacitance value measured with the parallel of ❶ and ❷. Finally, assuming that the sensing circuit has a resolution of 1 pF, it corresponds to 1.6% of the full-scale range. Such granularity is largely suitable for monitoring the filling status of the network segments.

### 3.5. Ionic Conductivity

The most interesting contactless measurement in this context is the estimation of water ionic conductivity. Conductivity is a fundamental parameter in monitoring water quality. As shown by simulations, by properly choosing the sensing frequency, it is possible to detect changes in water salinity. Here, in the absence of surrounding soil, we selected 20 MHz based on simulations of Figure 3a. Since we target variations of the value of R_W_ in a spectral region where the impedance is dominated by the capacitance (C_W_ in this case), and thus it is mostly imaginary, we leverage the phase sensitivity of the LIA and look at the real part of the impedance. In Figure 8, the time tracking of the real part of impedance at 20 MHz is reported for an experiment of increase in water salinity. The initial and final values of water conductivity are measured in the tank, in still conditions, with a handheld conductivity meter (PCE-PHD1 by PCE Instruments, Lucca, Italy) using a reference conductivity probe. Initially, the loop is filled with tap water with a conductivity of 700 μS/cm. Then, after four seconds, 50 g of salt (NaCl) is added to the tank. This amount of salt dissolves in the total volume of water (60 L, 30 in the tank to keep the pump immersed and 30 in the loop), increasing the conductivity to 1400 μS/cm. The curve shows an interesting transient: after 2 s from the injection of salt in the tank, a peak of conductivity is sensed. This is associated with the cluster of salt grains passing under the electrodes. When the water pushes away the salt, conductivity decreases again, and the regime value of conductivity is reached after 50 s. This time interval corresponds to about 10 rounds of water in the loop at a speed of 1.1 m/s, during which salt dissolves and oscillations smooth. 

Thanks to the insulation of the electrodes from the water, the sensor is very robust and stable from both the mechanical and chemical points of view. The resolution in the measurement depends on the noise of the readout instrument, which in this case was much below 0.1% of the measured impedance values. Since water conductivity also depends on its temperature (with a nonlinear change of ~2%/°C), these tests were performed with the pump running for several minutes in order to reach a stable temperature before salt injection. For an accurate measurement of conductivity, with an accuracy of ~1%, water temperature should be measured as well, with an accuracy better than 1 °C. In these experiments, we assumed a full filling condition of the pipe. In the case of a partial filling, the conductivity estimation should be corrected by the volume fraction, that can be estimated by the capacitive measurement illustrated in the previous section.

Finally, since we had glued two identical sets of electrodes in series along the pipe (❶ and ❷ in Figure 4), we also tested the possibility of estimating water speed by means of the delay in the propagation of the conductivity front upon injection of a small quantity of salt (2 g in this case). As visible in Figure 9, impedance is measured simultaneously on both channels (thanks to the dual input of the LIA). As expected, sensor two detects the front with a delay of ~10 ms, which is consistent with the water speed (~1 m/s) and the longitudinal distance between the electrode sets (10 mm).

### 3.6. Electrodes Manufacturing

In our tests, we employed copper electrodes (copper adhesive tape by 3 M and Advance Tapes, Thurmaston, UK) that can be easily cut into the desired shape and stuck on the pipes. However, since the metal is not in contact with the liquid and, thus, not exposed to electrochemical reactions, any metal could be used in this case. Even other conductive materials, such as polymers loaded with conductive particles, could be used, provided that the resistivity would not be too large to affect the measurement of R_W_. Copper is rather expensive, and consequently, more affordable metals could be employed. 

If the pipe is surrounded by soil, electrodes should be protected from corrosion. A straightforward solution is to cover the electrodes with a thin layer of Kapton. We used a standard adhesive Kapton foil with a thickness of 50 μm. No peel-off or damage of Kapton was observed during the experiments, confirming its robustness. The combination of copper and Kapton was also chosen to show the feasibility of adopting a standard industrial technology used to manufacture flexible printed circuit boards (PCB), which are now well consolidated, and also to manufacture strip electrodes. 

We performed a last test to compare experimentally the impedance measured with two pairs of electrodes of identical geometry, glued on the same 200 mm pipe but fabricated in different ways. One pair was made of copper tape manually covered by adhesive Kapton (50 μm) analogously to all the measurements reported here. The second pair was made from an industry-manufactured flexible PCB (with a Kapton thickness of 100 μm). As expected and shown in Figure 10, the measured spectra overlap consistently between the two setups when electrodes are surrounded by different media: air (pure capacitance), water (low resistance), and soil (where the CPE at low frequency differs due to the different Kapton thickness). Impedance spectra are measured up to 2 MHz with a precision LCR meter (E4980A by Agilent, Santa Clara, CA, USA). We can conclude that single-sided flexible PCBs can be used to quickly fabricate band electrodes to be applied to pipes. However, the typical maximum length offered by PCB manufacturers is ~1 m. Thus a special process should be put in place for the industrial production of electrodes embedded in longer pipes.

## 4. Discussion

The experimental results reported here show the versatility of contactless sensors for water monitoring through plastic pipes. PVC has enough rigidness to transmit ultrasounds and allow flow rate measurement with a resolution (a noise floor of ~0.6% of the full-scale range of 5 m/s) sufficient for network monitoring applications. Furthermore, we have shown that water temperature can be estimated with a sub-°C resolution with the same piezo transducers measuring the ToF of acoustic waves in water. However, for an accurate absolute value of temperature, an independent sensor of water pressure is needed. 

Metal electrodes applied along a section of the pipe enable a variety of measurements: water conductivity, pipe filling fraction, and potentially external leaks. As we have demonstrated, in principle, the flow rate could be estimated by means of impedance as well by injecting small amounts of salt (harmless for water purity) and measuring the transit time of the conductivity pulse by means of a couple of sets of electrodes placed in sequence at a short distance. On the one hand, this approach would allow measuring flow rates with simple external electrodes without the need for piezoelectric devices. On the other hand, the controlled injection of salt in the water would, of course, require an intrusive device, thus jeopardizing the advantages of a fully contactless approach.

This work targets primarily buried pipes, where temperature variations are very limited and slow. However, if surface pipes are considered, the effect and temperature and their daily variation have to be taken into account. Temperature affects the speed of sound in media, as we propose to leverage for approximate temperature sensing, but the measurement of the flow rate rejects this effect by computing the actual speed of sound in the water using the sum of the upstream and downstream waves. Impedance, instead, is directly affected by temperature variations: water conductivity by −2%/°C and its dielectric constant by −4%/°C. For accuracy in the measured impedance, both resistive (for conductivity) and capacitive (for filling level and void fraction), better than 10% is required, and then an independent temperature sensor should be added for continuous correction.

One of the main results of this work is the application of capacitive-coupled contactless conductivity sensing (often named C^4^D) to pipes with large diameters (here 90 mm), i.e., compatible with actual water distribution applications and high water speed, whereas in the literature only diameters below 10 mm and low flow rates were reported [28,29,30], mostly in microfluidics applications.

Finally, we can compare the capacitive sensitivity to a change of the medium inside the pipe (ΔC from air to water expressed in pF/m, i.e., normalized on the length of electrodes L) of the proposed configuration of electrodes with respect to the state of the art of capacitive sensing, for both two-phase flow identification in pipelines and level metering in tanks, summarized in Table 1. Again, it is apparent that our experimental demonstration refers to the largest diameter ever reported. Moreover, our electrodes are the longest among pipe sensing [31,32,33,34] and, despite the larger diameter, offer a good sensitivity (133 pF/m) in line with the best-performing ones. As discussed in the literature [31,33], guard electrodes can be flanked to the sensing electrode to improve the measurement accuracy by shaping a more uniform electric field and thus reducing the fringing and parasitic coupling effects at the edges.

## 5. Outlook

For this laboratory characterization, we have used commercial instruments, but we foresee the development of custom, miniaturized, and low-power electronics (similar to what was already performed for sensing nodes with biofilm monitoring [4]). From the point of view of the readout electronics, the system would be composed of four sections: (i) board control by a microcontroller and power management, (ii) ultrasounds section with pulse generation and echoes detection and timing, (iii) impedance section, with an LIA operating up to tens of MHz [38], and (iv) radio communication. In order to be energetically autonomous, the unit could harvest energy directly from the water flow, for instance, by means of a turbine [4], which is efficient but intrusive, or by means of a solar panel, which requires access to the surface.

## 6. Conclusions

We have presented the concept of a non-intrusive sensing node for smart monitoring of water with clamp-on sensors on pipes, piezoelectric transducers for ultrasound sensing, and copper electrodes for impedance. Pipes can be either accessible or underground, thanks to the small footprint of the sensors (0.5 m in our demonstrator). We have reported extensive experimental proofs of the feasibility of multi-parameter sensing, namely flow rate, temperature, conductivity, and filling fraction, in addition to the capability to detect leaks. The low cost of these sensors is crucial for water monitoring, differently from other application domains such as oil and gas, where the sensorization of the transport infrastructure can be more expensive. Moreover, it paves the way for their pervasive installation and consequently compensates for the lower performance with respect to state-of-the-art sensors. However, the typical requirements in terms of full-scale range, response time, resolution, and accuracy needed to monitor water distribution are achievable by the sensors here characterized. Furthermore, these technologies are compatible with other contactless sensing approaches, such as inductive ones [39] or magnetic transducers for communication along the pipes [40], as well as those inspection techniques based on electromagnetic waves of increasing frequency and penetration depth from THz to X-rays and gamma rays. The search for contactless sensing solutions includes pH as well. Although traditionally measured with electrochemical sensors, recent contactless alternatives leverage GHz resonators [41] or optical readout of pH patches (for instance, by PyroScience, Aachen, Germany) through transparent walls [42].

## Figures and Tables

**Figure 1 sensors-23-02075-f001:**
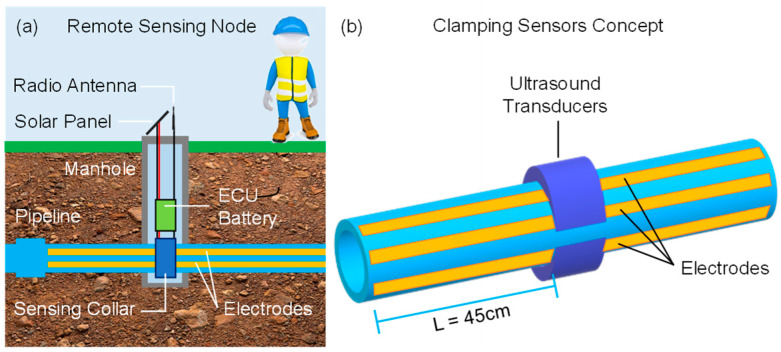
Envisioned smart water sensing node connected to an underground pipe (**a**) on which piezoelectric and impedance sensors are externally applied (**b**).

**Figure 2 sensors-23-02075-f002:**
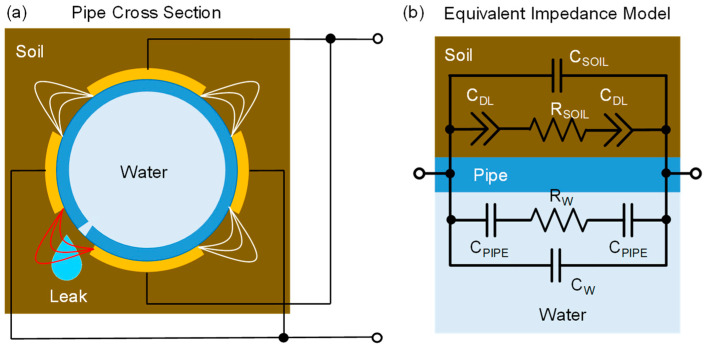
Cross section of the plastic pipe with external interdigitated electrodes (**a**) and corresponding equivalent impedance model (**b**) consisting of an inner and outer path.

**Figure 3 sensors-23-02075-f003:**
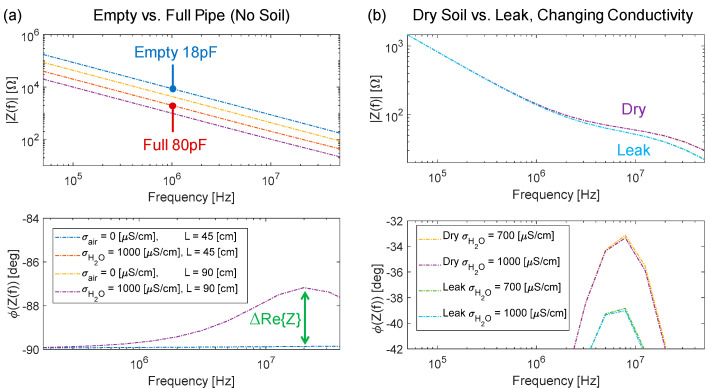
Bode plots (magnitude and phase) of the simulated impedance between the electrodes on the pipe in two conditions of surrounding materials: air (**a**) and soil (**b**).

**Figure 4 sensors-23-02075-f004:**
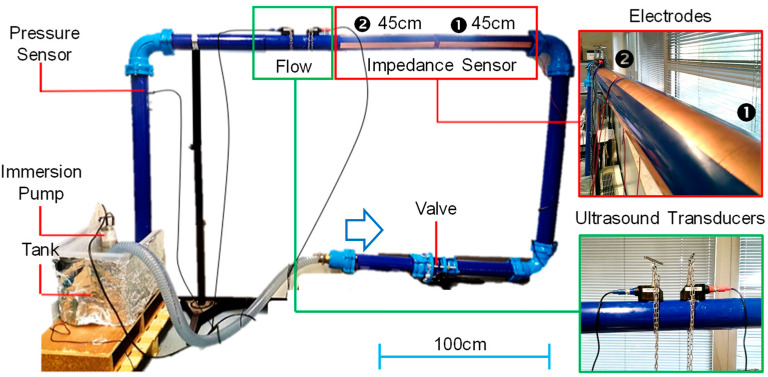
Hydraulic loop realized for the experimental validation of the contactless sensors at realistic water speed.

**Figure 5 sensors-23-02075-f005:**
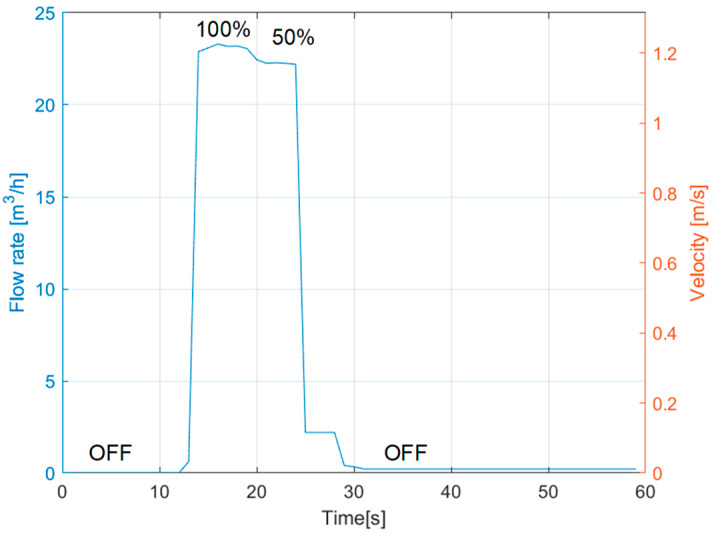
Ultrasonic measurement of water flow rate in the loop.

**Figure 6 sensors-23-02075-f006:**
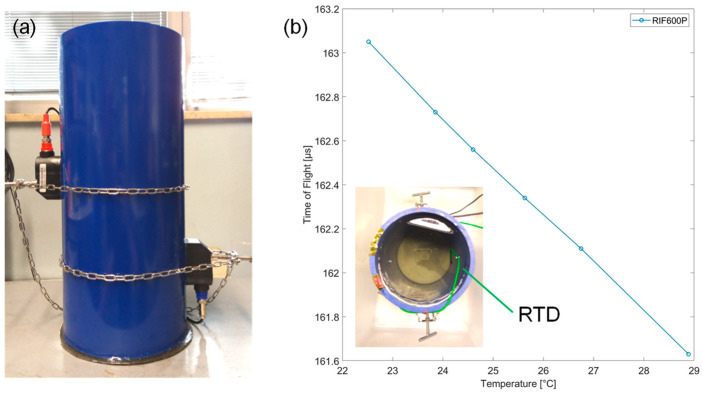
Water temperature measurement in a static setup (**a**) by means of the time of flight (**b**) of ultrasounds.

**Figure 7 sensors-23-02075-f007:**
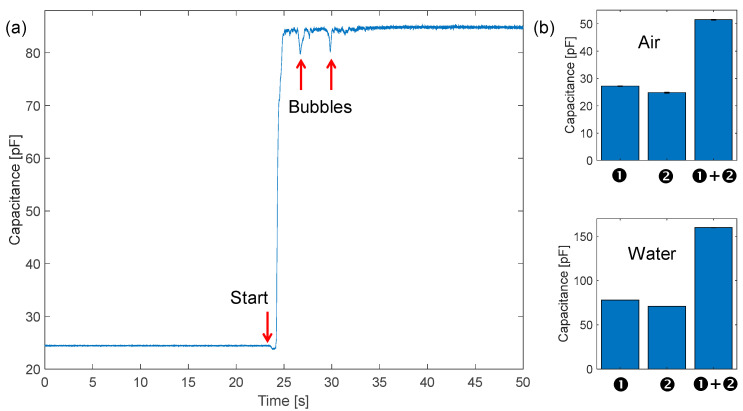
(**a**) Dynamic time tracking of capacitance at 1 MHz during the filling transient of the pipe. (**b**) Static capacitance measurements with two sets of electrodes (45 cm each), connected individually or in parallel (90 cm ❶ + ❷) doubling the signal in two static cases: empty (air) and full (water).

**Figure 8 sensors-23-02075-f008:**
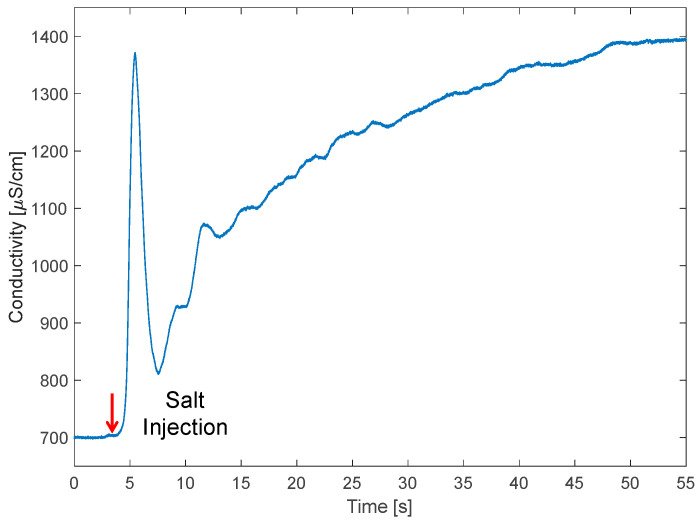
Time tracking of the real part of impedance at 20 MHz during a conductivity increase produced by the injection of 50 g of NaCl in the tank.

**Figure 9 sensors-23-02075-f009:**
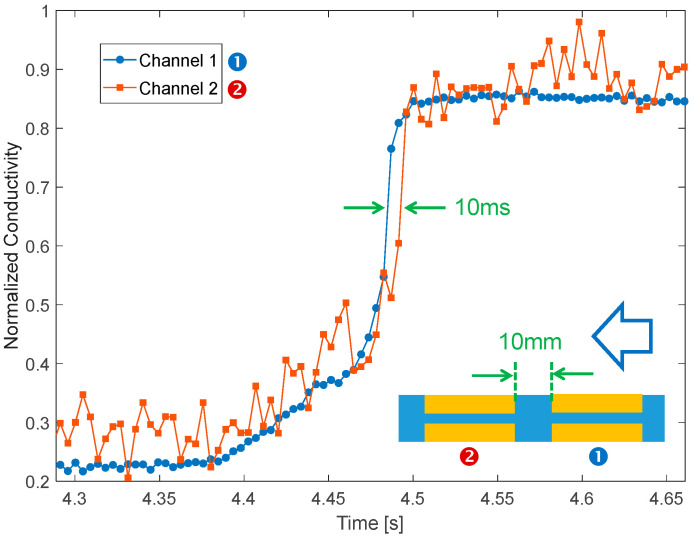
Dual channel conductivity tracking upon injection of 2 g of NaCl showing potential for velocimetry. The arrow indicates the water direction.

**Figure 10 sensors-23-02075-f010:**
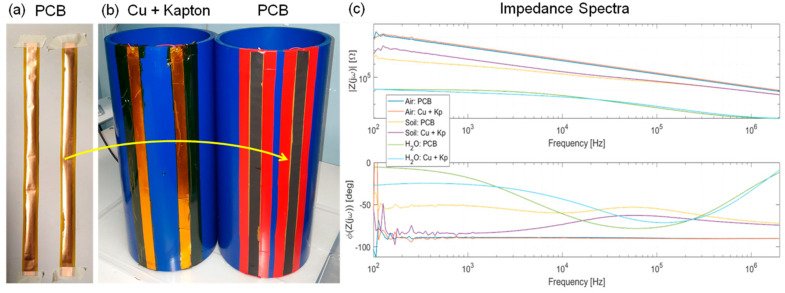
Comparison of the copper electrodes fabrication approaches manual vs. industrial flexible PCB manufacturing: the measured impedance spectra overlaps in all conditions (air, water and soil) demonstrating their equivalence.

**Table 1 sensors-23-02075-t001:** Comparison of contactless capacitive sensors applied to pipes and tanks.

Application	Geometry	Diam. [mm]	Length [cm]	ΔC/L [pF/m]	Frequen. [kHz]	Ref.
Filling status	Longitudinal	90	45	133	1000	This
Two-phase flow	Longitudinal *	25	10	500	2000	[31]
Void fraction	Longitudinal	50	12.5	12	N.A.	[32]
Slug flow	Longitudinal *	34	0.5	140	N.A.	[33,34]
Level	Cylindrical	26	600	300	32	[35]
Level	IDE	-	4	40	25	[36]
Level	Coil	33	40	400	1	[37]

* Sensing electrode surrounded by guard electrodes.

## Data Availability

Data are available upon reasonable request.

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
