# Peer review of "Contactless Sensing of Water Properties for Smart Monitoring of Pipelines"

_sensors, 2023, doi:10.3390/s23042075_

Round 1
Reviewer 1 Report
This paper is well written. However, it is important to address the following queries to enhance the quality of the manuscript.
1) Authors needs to justify the novelty of the proposed work in tabular form and include them in the manuscript.
2) Since the application demonstrated in this work can be used further for the field measurements it is important to examine the effect of diurnal temperature variations on the sensor as well as on the system and include it in the manuscript.
3) In section 3.6, 3rd paragraph authors have mentioned that “. One pair was made of copper tape covered by Kapton (50 μm) analogously to 328 all the measurements here reported.” Did author notice any peel of the kapton during the measurements, which I believe is quite obvious.
4) Authors needs to demonstrate the stability of the sensors used for Ionic Conductivity.
5) The quality/resolution of the Fig. 10 (c) and (d) are very poor. Please improve it.
6) In section 3.2, authors have mentioned “It is negligible in our case where accuracy of a few % is acceptable.” What is this ‘’few” referred to? Please provide the both quantitative and qualitative information.
Author Response
See attached file with detaile response and list of modifications.

Reviewer 2 Report
The paper discussed two types of sensors: piezoelectric sensors and strip electrodes, for water monitoring. The comments are listed below:
1. The technique of using an ultrasonic sensor for flow measurement and water temperature does not show anything new. The ultrasonic flow meter is a mature sensor with commercial products available. If the authors want to integrate this into the water monitoring system, low-cost PZT plates might be a better option rather than commercial ultrasonic sensors.
2. Additionally, using the speed of sound in water for water temperature monitoring should be used carefully. First, both the temperature change of the water and pipe will affect the final ToF. A temperature gradient will exist if the pipe is buried, and the gradient will cause large measurement errors. This might be a good idea for temperature measurement, but the measurement system should be carefully calibrated.
3. It is an interesting idea to use capacitance to measure the filling status. However, in this work, more data is needed to validate the method, such as using multiple filling depths rather than using only two: empty and full. In the measurement, another method should be used as a comparison.
4. Same for the ionic conductivity measurement: another method is needed for quantitative comparison to validate the accuracy of the method used here.
5. In section 3.6, the authors should furtherly analyze the results in Figure 10 and give a conclusion, such as which setup is the best or how to improve the current setup.
Author Response

(The authors gave the same response as above.)

Reviewer 3 Report
I want to congratulate the authors for their efforts in the manuscript. The authors have presented a paper in which an inductive sensor is presented for water quality monitoring in grids. The paper is, in general terms, well presented. Nonetheless, there are some problems that authors need to solve before accepting the paper for publication. Following, I include a list of these problems Section by Section.
The abstract should be extended. Add more information on the used sensors and performed experiments.
Keywords: consider adding grid monitoring
Section 1: More references are needed to justify the information provided in the introduction. In addition, the aim and structure of the paper must be added. For the aim of the paper, I suggest using a paragraph with bullet points to highlight the most relevant aspects. For the structure, add a short paragraph at the end of the introduction defining the content of each one of the subsequent sections.
Related work/State of the art: This section (or subsection) is missing. The authors need to add a section after the introduction in which they show the existing solutions for water quality sensing in the grid and the contactless sensors. Following, I added some papers aligned with their proposal (10.1109/AICCSA.2018.8612845, 10.1016/j.measurement.2019.107260, 10.2166/ws.2021.342).
Section 2: Add more examples of inductive sensors for water quality. Some examples of water conductivity and water level can be found in the literature.
Section 3: Some Figures need to be reshaped to ensure that they are readable in printed versions. Consider modifying the letter size and colours of lines and use dotted and striped lines.
Section 4 must be extended. This is mandatory. The comparison of results with other work, the identification of the impact of the proposed system on water grids and the limitations of experiments must be explained.
Section 5: Please add the future work in a new paragraph. Consider moving some of this content of Section 4 to ensure that the conclusions are short.
Author Response

(The authors gave the same response as above.)

Round 2
Reviewer 2 Report
The reviewer does not have any further comments, and the paper can be accepted in the present form.
Author Response
We thank the reviewer for proposing to accept the paper in the current, revised form.
Reviewer 3 Report
The authors have addressed most of my comments. Nonetheless, I still have some suggestions:
The abstract should be enlarged a bit more.
The novelty of the paper must be described with more detail and the paragraph in which it is described should be moved before the structure of the paper.
Author Response
See attached pdf file.
